# Lymphocyte Population Changes at Two Time Points during the Acute Period of COVID-19 Infection

**DOI:** 10.3390/jcm11154306

**Published:** 2022-07-25

**Authors:** Giulia Scalia, Maddalena Raia, Monica Gelzo, Sara Cacciapuoti, Annunziata De Rosa, Biagio Pinchera, Riccardo Scotto, Lorella Tripodi, Mauro Mormile, Gabriella Fabbrocini, Ivan Gentile, Roberto Parrella, Giuseppe Castaldo, Filippo Scialò

**Affiliations:** 1CEINGE-Biotecnologie Avanzate, Scarl, 80131 Naples, Italy; scalia@ceinge.unina.it (G.S.); raia@ceinge.unina.it (M.R.); monica.gelzo@unina.it (M.G.); tripodi@ceinge.unina.it (L.T.); filippo.scialo@unicampania.it (F.S.); 2Dipartimento di Medicina Molecolare e Biotecnologie Mediche, Università di Napoli Federico II, 80131 Naples, Italy; 3Dipartimento di Medicina Clinica e Chirurgia, Università di Napoli Federico II, 80131 Naples, Italy; sara.cacciapuoti@libero.it (S.C.); biapin89@virgilio.it (B.P.); riccardo.scotto@unina.it (R.S.); mormile@unina.it (M.M.); gafabbro@unina.it (G.F.); ivan.gentile@unina.it (I.G.); 4Dipartimento di Malattie Infettive e Emergenze Infettive, Divisione di Malattie Infettive Respiratorie, Ospedale Cotugno, AORN dei Colli, 80131 Naples, Italy; annunziataderosa@yahoo.it (A.D.R.); rob.parrella@gmail.com (R.P.); 5Dipartimento di Medicina Traslazionale, Università della Campania L. Vanvitelli, 80131 Naples, Italy

**Keywords:** COVID-19, lymphocyte subpopulations, serum interleukins, steroid therapy

## Abstract

We previously observed an increase of serum interleukins (IL) and a reduction of most lymphocyte subpopulations in hospitalized COVID-19 patients. Herein, we aimed to evaluate the changes in serum IL-6, IL-10, and IL-17A levels and cytometric lymphocyte profiles in 144 COVID-19 patients at admission and after one week, also in relation to steroid treatment before hospitalization. After one week of hospitalization, we found that: (i) total lymphocytes were increased in all patients; (ii) neutrophils and IL-6 were reduced in mild/moderate patients; (iii) B lymphocytes were increased in severe patients; (iv) T lymphocyte populations increased in mild/moderate patients. In the eight patients that died during hospitalization, total leukocytes increased while T, T helper, T cytotoxic, T regulatory, and NK lymphocytes showed a reducing trend in five of the eight patients. Even if seven days are too few to evaluate the adaptive immunity of patients, we found that the steroid therapy was associated with a reduced COVID-19 inflammation and cytokine activation only in patients with severe disease, while in patients with less severe disease, the steroid therapy seems to have immunosuppressive effects on lymphocyte populations, and this could hamper the antiviral response. A better knowledge of cytokine and lymphocyte alterations in each COVID-19 patient could be useful to plan better treatment with steroids or cytokine targeting.

## 1. Introduction

COVID-19 infection is associated with systemic inflammation [1] that involves mainly lung endothelium, which is particularly rich in the ACE2 receptor [2]. Severe respiratory involvement appears in 5–20% of cases with pneumonia that may progress to acute respiratory distress syndrome (ARDS) and multiorgan failure [3]. A hallmark of COVID-19 is lymphopenia which is related to the severity of the disease. The T response is mainly involved in mechanisms that include defective T cell proliferation and enhanced apoptosis. In particular, T helper lymphocytes are more susceptible to apoptosis, while cytotoxic lymphocyte reduction is due to their exhaustion [4] due to infiltrates and sequestration, particularly in the lung. Lymphopenia is mainly triggered by the enhanced production of proinflammatory cytokines, among which are interleukin (IL)-6 and IL-17 [5]. Furthermore, in COVID-19 patients, IL-6 inhibits the immune cell cytotoxicity [6,7]. Circulating natural killer (NK) lymphocytes are reduced in turn in severe COVID-19 patients with an altered balance and regulation of NK subsets [8].

The COVID-19 pandemic had two waves in Italy. During the first wave, which occurred between March and May 2020, we studied 35 COVID-19 patients at hospital admission [9] demonstrating that serum IL-6 was increased in all cases with a significant relationship with disease severity. The same patients showed severe lymphopenia that involved both B and T lymphocytes, which were also related to the disease severity. After a lockdown during the summer of 2020, the pandemic had a second wave in September 2020. The two waves had several differences [10]. In fact, during the first wave most patients were diagnosed after the onset of symptoms by molecular analysis on nasopharyngeal swabs, and severe patients were hospitalized soon after the result. Conversely, most patients from the second wave were diagnosed when they were still asymptomatic, mostly because they have had contact with a COVID patient and have been traced. The result of the nasopharyngeal test was obtained more rapidly, often before the onset of severe symptoms, and most patients started to be treated with combinations of dexamethasone and azithromycin. During the second wave, we studied 170 COVID-19 patients at hospital admission, demonstrating that such patients had lower serum IL-6 levels and less severe inflammation [11] and endothelial damage [12] as compared to patients of the first wave. Surprisingly, milder patients of the second wave had more severe lymphopenia (possibly induced by pre-hospitalization steroid treatment), while patients in advanced stages had less severe B and T lymphopenia as compared with patients of the first wave [13].

A better knowledge of cytokine and lymphocyte alterations in COVID-19 patients is important, since steroid drugs and the targeting of cytokines may be a relevant therapeutic option in these patients [14]. In the present study we evaluated the changes in serum IL-6, IL-10, and IL-17A levels and the cytometric lymphocyte profiles in 144 COVID-19 patients at admission and after one week of hospitalization, also in relation to steroid treatment before hospitalization.

## 2. Materials and Methods

### 2.1. Patients

We studied 144 adult patients with the diagnosis of COVID-19 admitted to one of our hospitals. The flow-chart of patients’ selection is reported in Appendix A. The study was approved by the Ethical Committee of the University Federico II of Naples (protocol code 138/20, 14 April 2020). All procedures have been carried out in accordance with the Declaration of Helsinki. Informed consent was obtained from all subjects; the lone exclusion criterion was the refusal or the impossibility to undersign the informed consent. The diagnosis of COVID-19 infection was confirmed by molecular analysis on nasopharyngeal swab [15], while patient’s severity was based according to WHO classification. Specifically, mild is defined as respiratory symptoms without evidence of pneumonia or hypoxia; moderate is defined as presence of clinical and radiological evidence of pneumonia with SpO2 ≥ 90% on room air; and severe is defined as presence of clinical and radiological evidence of pneumonia and a respiratory rate > 30 breaths/min or SpO2 < 90% on room air (World Health Organization. Clinical Management of COVID-19: Interim Guidance. World Health Organization; 2020:13–15.).

### 2.2. Lymphocyte Population and Serum Cytokines Analyses

Whole blood was collected at admission and after one week of hospitalization in tubes containing EDTA and then immediately analyzed by flow cytometry, and in tubes without anticoagulant for cytokine analysis. Cytometric analysis was performed by multicolor flow cytometry (Facs Canto II; Becton Dickinson Italia, Milan, Italy) as previously described [9]. The gating strategy was as follows: (A) lymphocyte cells were gated by using CD45 vs. SSC-A, and this gate was used to identify T lymphocytes (CD45+, CD3+). T helper (TH) and T cytotoxic cells have been identified as CD3+, CD4+ and CD3+, CD8+, respectively. Naïve and memory cells have been identified as CD45RA+ and CD45RO+, respectively. T regulatory cells have been identified as CD3+, CD4+, CD25+, CD127−. (B) From the lymphocytes (CD45 vs. SSC-A), activated T lymphocytes have been identified as CD3+, DR+. (C) From TH cells (CD3+, CD4+), TH17 and TH1 have been identified as CCR6+, CXCR3− and CCR6−, CXCR3+, respectively. CD38+ and HLA-DR+ were used to identify their activated form. CD19 and CD56 were used to identify B cells and natural killer (NK) cells respectively (Appendix A). Facs Diva software was used to analyze the collected cells. The values have been expressed both as a percentage and absolute numbers.

Serum IL-6, IL-10, and IL-17A levels were analyzed using ELISA Max™ Set Deluxe kits (BioLegend, Inc., San Diego, CA, USA), in accordance with the manufacturer’s instructions [9]. For 24 COVID-19 patients, we also collected a sample one month after hospitalization.

Serum ferritin and C-reactive protein were analyzed by commercial kits (Abbott Diagnostics, Rome, Italy) on an automated biochemistry analyzer (Architect ci 16200 Integrated System, Abbott Diagnostics, Rome, Italy).

### 2.3. Statistical Analysis

Non-parametric continuous data were reported as median (interquartile range). Shapiro-Wilk test was used to test the normality of distributions. Δ (%) has been calculated as follows: (after 1 week-basal)/basal × 100. Comparisons between two groups were evaluated by Mann–Whitney U test. Paired comparisons between the data at hospital admission and after 1 week were evaluated by Wilcoxon signed-rank test. Categorical data were reported as frequency and percentage. The chi-square test was used to compare the frequency of categorical variables. Statistical analysis was performed by SPSS (version 27, IBM SPSS Statistics, Armonk, NY, USA). Graphics have been performed by KaleidaGraph software (version 4.5.4, Synergy, Reading, PA, USA) and Excel (Office 365, Microsoft, Redmond, WA, USA). *p* values < 0.05 were considered as significant.

## 3. Results

We studied 144 patients classified based on WHO classification described in material and methods. Specifically, they have been classified as mild (*n* = 29), moderate (*n* = 79), and severe (*n* = 28) on the basis of clinical symptoms reported in Table 1. Among the 144 patients, 8 (5.5%) patients died from COVID-19 at an average of 11 days from hospitalization (range: 7–23 days). Among the 144 patients, 80 (55%) were treated with dexamethasone before hospitalization for 2–3 days, while 64 (45%) were not treated. During hospitalization, the treatment varied according to severity and responsiveness to ongoing steroid therapy. In case of severe disease evolution, immunomodulators (such as tocilizumab) were administered and/or the dosage of steroids was increased. In Table 1 we report the demographic data, clinical symptoms, and hematological profile of mild, moderate, severe, and patients who died both at admission and after one week of hospitalization.

The median age of mild patients was significantly lower than moderate, severe, and dead patients (*p* < 0.001), while no differences were observed among the other three groups. Regarding gender, we found a significant difference only between mild and dead patients (*p* = 0.047). We observed a significant reduction of patients with fever and cough in all four groups, while the number of patients that needed respirators and intubation increased significantly in moderate, severe, and dead groups after one week of hospitalization. Moderate and severe patients showed a significant reduction of platelets (moderate: basal median 356 (range 278–456), 1 week median 330 (range 287–382) *p* < 0.0001; severe: basal median 432 (range 318–457), 1 week median 375 (range 319–423) *p* < 0.005).

Neutrophils were significantly reduced as a percentage in mild and moderate patients (mild: basal median 70 (range 60–75), 1 week median 66 (range 50–72) *p* < 0.030; moderate: basal median 77 (range 71–82), 1 week median 73 (range 60–79) *p* < 0.0001), while the absolute number was significantly increased in patients who later died (basal median 3315 (range 2721–5585), 1 week median 7560 (range 3758–12,830). The number of total lymphocytes was significantly higher in patients of all WHO subgroups (mild: basal median 1160 (range 621–1858), 1 week median 1663 (range 1195–2185) *p* < 0.03; moderate: basal median 988 (range 614–1500), 1 week median 1256 (range 904–1819) *p* < 0.0001; severe: basal median 991 (range 634–1783), 1 week median 982 (range 350–1340), except for patients who died. The neutrophil/lymphocyte ratio (NLR) was significantly reduced in mild and moderate patients (mild: basal median 6.3 (range 3.8–10), 1 week median 3 (range 1.7–4) *p* < 0.023; moderate: basal median 6.9 (range 3.5–9.7), 1 week median 4.3 (range 2.5–6.6) *p* < 0.001), while it showed an increasing trend in patients who died (died: basal median 4.3 (range 2.8–12), 1 week median 8.8 (range 5.1–20) *p* = ns). Thus, we evaluated the NLR at admission (basal), after one week of hospitalization and after one month of hospitalization in 24/144 COVID-19 patients. As shown in Figure 1 the NLR showed a reducing trend after 1 week and after 1 month in mild patients (panel A), while it was significantly reduced after one week of hospitalization and further significantly reduced after one month in moderate patients, as compared to the basal value (panel B). Among severe patients (panel C), we could evaluate NLR after 1 month in only two patients that showed variable trends.

After one week of hospitalization, the serum levels of ferritin and C-reactive protein were significantly reduced in mild, moderate, and severe patients, while no significant differences were observed in patients who later died (mild: basal median 19 (range 12–46), 1 week median 5 (range 4–10) *p* < 0.0001; moderate: basal median 45 (range 25–68), 1 week median 11 (range 6–18) *p* < 0.0001; severe: basal median 63 (range 36–78), 1 week median 19 (range 9.5–61) *p* = 0050).

Table 2 shows the data of serum IL-6, IL-10, and IL-17A in 144 COVID-19 patients at hospital admission in comparison with those obtained after one week of hospitalization. After one week of hospitalization, we observed that: serum IL-6 was significantly lower in mild and moderate patients (mild: basal median 40 (range 26–129), 1 week median 30 (range 23–66) *p* < 0.031; moderate: basal median 133 (range 37–275), 1 week median 59 (range 24–170) *p* < 0.043), while no significant changes were observed in severe patients and those who later died. Serum IL-10 was significantly lower only in moderate patients (moderate: basal median 9.3 (range 5.4–19.4), 1 week median 6.7 (range 4.6–9.2) *p* < 0.002). Serum IL-17A was not significantly different after one week of hospitalization in none of the four subgroups of patients.

Figure 2 shows the numbers of other lymphocyte populations in the 144 COVID-19 patients at admission in comparison with those obtained after one week of hospitalization. After one week of hospitalization, we observed that the percentage and the absolute number of B lymphocytes were significantly higher in dying and severe patients (severe: basal median 125 (range 65–247), 1 week median 150 (range 82–302) *p* < 0.021); died: basal median 9.5 (range 3–17), 1 week median 14 (range 4.2–26) *p* < 0.034), respectively, while no significant change was observed in mild and moderate patients (Appendix A). The number of naïve results significantly increased in mild and moderate patients (mild: basal median 655 (range 459–1207), 1 week median 973 (range 681–1492) *p* < 0.012); moderate: basal median 571 (range 411–977), 1 week median 764 (range 474–1242) *p* < 0.003); the number of NK was significantly increased only in mild patients (mild: basal median 103 (range 45–196), 1 week median 156 (range 69–227) *p* < 0.016), while the percentage of NK was significantly reduced in moderate patients (moderate: basal median 10 (range 5–15), 1 week median 8 (range 5–13) *p* < 0.016). The number of memory lymphocytes was increased in all subgroups of patients, except for dying patients that showed a decreasing trend (mild: basal median 477 (range 192–718), 1 week median 618 (range 425–806) *p* < 0.001; moderate: basal median 354 (range 202–564), 1 week median 474 (range 346–651) *p* < 0.0001; severe: basal median 350 (range 214–509), 1 week median 534 (range 306–767) *p* = 0.02; died: basal median 333 (range 161–683), 1 week median 281 (range 130–534).

Figure 3 shows the numbers of activated and TH lymphocyte populations in the 144 COVID-19 patients at admission in comparison with those obtained after one week of hospitalization. After one week of hospitalization, we observed that: the number of total activated, and total T activated was significantly increased after one week of hospitalization in moderate and/or severe patients (Total activated, moderate: basal median 173 (range 101–286), 1 week median 204 (range 123–377) *p* < 0.020; severe: basal median 169 (range 102–281), 1 week median 184 (range 127–342) *p* = 0.021; T activated, moderate: basal median 23.7 (range 12.9–34.5), 1 week median 32.8 (range 16.6–62.4) *p* < 0.022) (Appendix A). The number of total TH1 lymphocytes was significantly increased in mild and moderate patients (mild: basal median 147 (range 56–180), 1 week median 186 (range 99–244) *p* < 0.004; moderate: basal median 70 (range 43–133), 1 week median 119 (range 71–195) *p* < 0.0001), while the number of total TH17 was significantly increased only in severe patients (severe: basal median 29 (range 12–78), 1 week median 55 (range 38–81) *p* = 0.001).

A total of 80/144 (55%) patients were treated with steroids before hospitalization. Thus, we examined the data of cytokines and lymphocyte populations at hospitalization and after one week in the different severity subgroups of patients comparing treated and untreated patients. No significant differences were observed for any of the parameters between treated and untreated patients of the mild and moderate subgroups. While for severe and dying patients we observed several differences between the 24 treated and the 11 untreated patients. In fact, at hospital admission, treated patients had significantly lower levels of IL-6 and IL-10 as compared to untreated patients (IL6, not treated median 42.2 (range 31.2–403), treated median 24.3 (range (20–34.4) *p* < 0.008; IL10, not treated median 9.7 (range 4.8–57.1), treated median 1.7 (range 0.1–8.2) *p* = 0.015) (Table 3). In addition, after one week of hospitalization, we observed in treated patients a significant increase in total (basal median 169 (range 119–358), 1 week median 213 (range 136–349) *p* < 0.005), memory (basal median 350 (range 230–693), 1 week median 535 (range 324–733) *p* < 0.042), T cytotoxic (basal median 234 (range 129–405), 1 week median 328 (range 239–517) *p* < 0.009), B (basal median 125 (range 68–265), 1 week median 154 (range 98–303) *p* < 0.008), total activated (basal median 169 (range 119–358), 1 week median 213 (range 136–349) *p* < 0.005), T activated (basal median 17.1 (range 7.9–40.3), 1 week median 31.7 (range 17.3–57.7) *p* < 0.028), and TH17 lymphocytes (basal median 32 (range 19–68), 1 week median 71 (range 44–86) *p* < 0.0001) as compared to untreated patients, while no significant differences were observed in untreated patients.

Figure 4 shows that in 6/8 patients the number of total leukocytes was increased after one week of hospitalization while, in 5/8 the number of T lymphocytes, T helper, T cytotoxic, T regulatory, and NK lymphocytes, showed a trend of reduction after one week of hospitalization. In all the 8 patients the number of total activated and TH1 activated lymphocytes was increased after one week of hospitalization (data not shown).

## 4. Discussion

Since the beginning of this pandemic, the immune response to COVID-19 infection has been the focus of many studies where an increase in cytokine levels and lymphopenia have been found to be a common characteristic and correlated with severity. Two interesting studies have shown that COVID-19-severe patients present with an immunological feature that are both shared and different from mild and asymptomatic patients [16] and provided a single cell transcriptome lung atlas from lung of dead COVID-19 patients unraveling the pathological pathway driving severe phenotype [17]. We have previously observed the increase of serum ILs and the reduction of most lymphocyte populations in blood from hospitalized COVID-19 patients, either during the first [9] and during the second [13] pandemic waves in Italy. These alterations correlated with the severity of the disease, in agreement with the current literature [6,8,18,19,20,21,22]. In the present study, we compare markers of inflammation and lymphocytes population at the time of hospitalization and after 7 days demonstrating that most of such alterations are transient, and after one week of hospitalization, they tend to normalize, although with some differences between patients with different severity stages.

In fact, total circulating lymphocytes were increased after one week of hospitalization in all subgroups of patients, while neutrophils were slightly reduced. The increase of lymphocytes may depend on a restored production since they were increased both as a percentage of total blood cells and as an absolute number, and because the number of naïve lymphocytes was increased in all severity subgroups of patients. This normalization of cell counts, already significant after one week of hospitalization, is maintained in the time since the NLR [23], which is significantly decreased after one week of hospitalization, further decreased after one month.

In addition, both serum IL-6 and IL-10 significantly declined after one week of hospitalization in mild and moderate COVID-19 patients. This suggests that the release of these molecules during COVID-19 infection is a transient event—at least in less severe patients—and agrees with the rapid kinetics of these serum biomarkers observed in preclinical models [24] and in other human diseases [25]. While, both IL-6 and IL-10, as well as serum IL-17A, showed an increasing trend in severe COVID-19 patients after one week of hospitalization suggesting that, in more severe patients, the inflammatory status and the release of cytokines is a more prolonged phenomenon [26,27]. NK cells, which in turn are inhibited by cytokines [20], were also reduced in most patients at hospital admission and increased after one week of hospitalization only in mild patients. It is known that NK are mainly involved in the primary control during COVID-19 infection [6]. Thus, the reduction of this population and the lack of a rapid cell count normalization in severe patients should be further studied in order to define specific treatments. Furthermore, recent studies are showing that NK cells from COVID-19 severe patients had a reduced cell-cell interaction and degranulation possibly caused by an early expression of TGFB [28] and a dysregulated IFN-α and TNF response [29].

Interestingly, B cells were reduced in most COVID-19 patients at hospitalization and increased significantly after one week of hospitalization in patients who later died as percentage and in severe patients as an absolute number. This finding is novel since most previous studies reported that B lymphocytes are less involved than T lymphocytes in COVID-19 [3]. The mechanisms that cause B lymphopenia in COVID-19 patients are still unknown, but the cell count normalization that we observed particularly in patients of advanced stages may depend on the stimulating effect of cytokines on lymphocyte B [30].

Finally, activated lymphocytes increased after one week of hospitalization mainly in severe/dying patients. These data match with the trend of serum interleukins, indicating a vicious circle in severe COVID-19 patients between cytokine production and TH lymphocyte activation, which slows down disease remission [31] Interestingly, steroid therapy performed before the hospitalization seem to not influence the levels of serum cytokines and lymphocyte population profiles in mild/moderate patients, while in severe patients it is associated with lower levels of cytokines at hospital admission and the increase of lymphocyte population cell counts after one week of hospitalization. This finding suggests that these cytokines may have a relevant role in causing the lymphopenia observed in COVID-19 patients enhancing lymphocyte apoptosis and inhibiting lymphocyte maturation [4]. Steroid therapy seems to reduce the cytokine activation only in patients with severe disease, consequently limiting their inhibitory effects on T lymphocyte cell count [32], while in patients with less severe disease (not requiring oxygen supplementation) it seems that steroid therapy may display immunosuppressive effects on lymphocyte populations, hampering the antiviral response [33,34]. In any case, even if the steroid treatment is associated with the reduction of ILs levels and the normalization of the number of lymphocytes in patients of advanced stages, most likely it is not sufficient to control the inflammation, as in these patients we observed an increase of T activated populations after one week of hospitalization. This suggests that in more severe COVID-19 patients with higher serum values of IL-17A and/or a higher number of circulating activated TH17, this pathway should be specifically targeted, as we suggested in a previous study [9].

A limitation of this study is represented by the limited period of follow-up of one week. We selected this time point as it coincided with the de-hospitalization of most patients, in particular mild patients, and recalling the patient for the follow-up during the pandemic has been difficult. Further studies with a prolonged follow-up period of COVID-19 patients need to confirm these preliminary results. Another study limitation is that the basal time should be defined for the period after SARS-CoV-2 RT-PCR positivity, while for this study the basal time corresponds to the hospitalization.

## 5. Conclusions

Here we compare markers of inflammation and lymphocytes population in patients at the time of hospitalization and after 1 week of hospitalization. Although seven days are few to evaluate the adaptive immunity of patients, our study shows that in less severe hospitalized COVID-19 patients there is a rapid normalization of the cytokine levels and the lymphocyte cell counts. In more severe cases the normalization is less significant, and in patients who died of COVID-19 the cytokine storm and the consequent lymphopenia became more pronounced after one week, suggesting the targeting of specific IL pathways. In this context, immunocytometric biomarkers may help clinicians to select the optimal treatment for each patient.

## Figures and Tables

**Figure 1 jcm-11-04306-f001:**
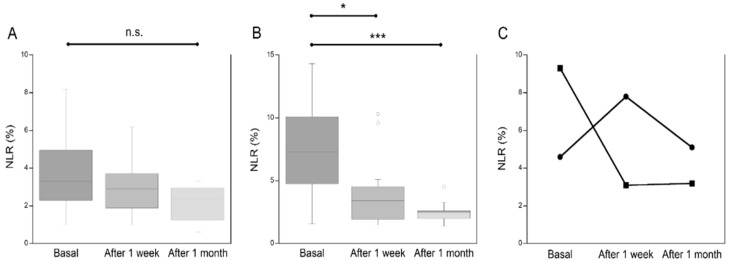
Neutrophil to lymphocyte ratio (NLR) in mild ((**A**); *n* = 7), moderate ((**B**); *n* = 15) and severe ((**C**); *n* = 2) COVID-19 patients at hospital admission (basal), after one week and after one month of hospitalization. * *p* < 0.01; *** *p* < 0.0001; n.s.: not significant.

**Figure 2 jcm-11-04306-f002:**
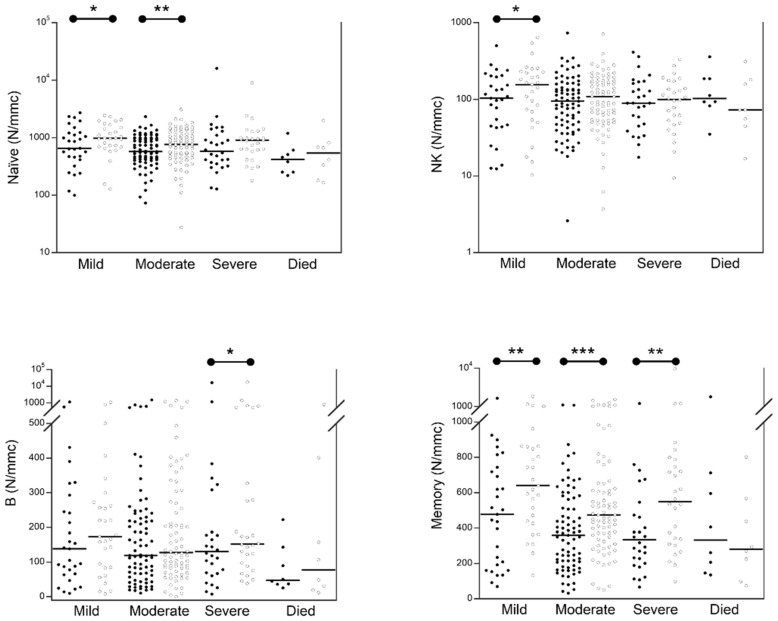
Comparison of circulating lymphocyte subpopulations in 144 COVID-19 patients with different severity at hospital admission (black circle) and after 1 week (white circle). The black lines represent the median values. * *p* < 0.05; ** *p* < 0.005; *** *p* < 0.0005.

**Figure 3 jcm-11-04306-f003:**
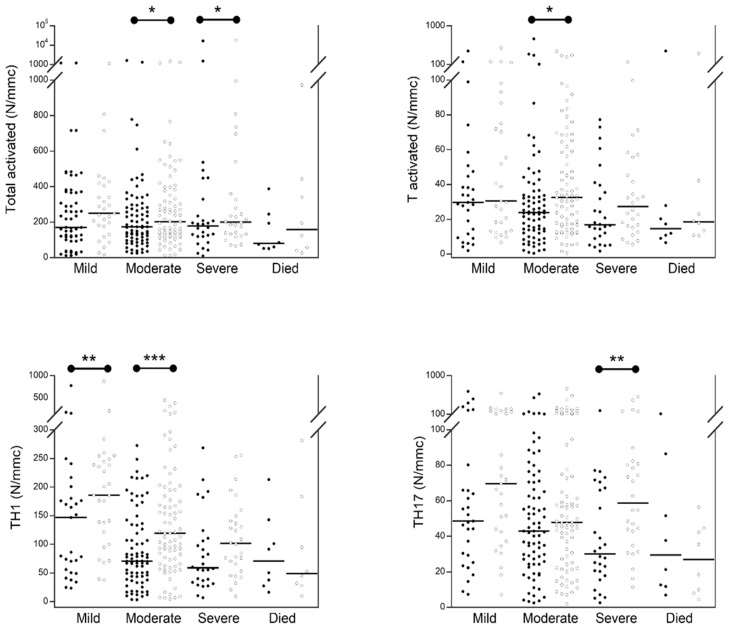
Comparison of activated and TH lymphocyte populations in 144 COVID-19 patients with different severity at hospital admission (black circle) and after 1 week (white circle). The black lines represent the median values. * *p* < 0.05; ** *p* < 0.005; *** *p* < 0.0005.

**Figure 4 jcm-11-04306-f004:**
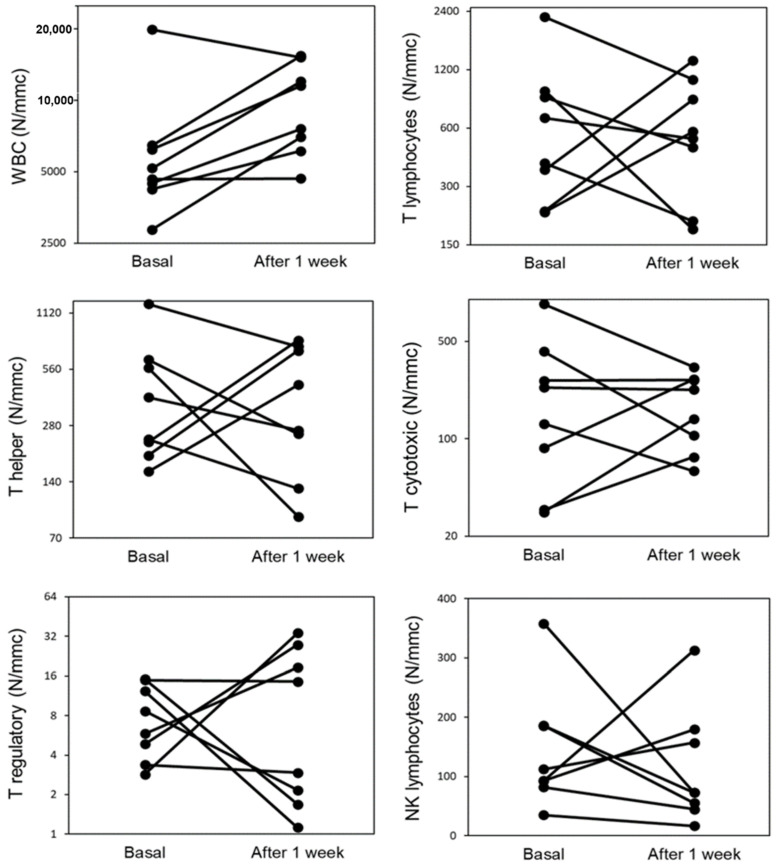
Circulating leukocytes (WBC), T and NK lymphocyte populations in COVID-19 patients that died during hospitalization (*n* = 8), at hospital admission (basal) and after one week of hospitalization.

**Table 1 jcm-11-04306-t001:** Patient characteristics.

		Mild (*n* = 29)	Moderate (*n* = 79)	Severe (*n* = 28)	Died (*n* = 8)
Age (years)		40 (30; 61)	61 (50; 71)	62 (52; 72)	78 (58; 85)
Males (*n*, %)		14 (48)	50 (63)	19 (68)	7 (87)
Oxygen	Basal	16 (55)	66 (83)	24 (86)	7 (87)
support (*n*, %)	After 1 week	12 (41)	64 (81)	23 (82)	6 (75)
	Δ (%)	−25	−3.0	−4.2	−14
	*p* value	n.s.	n.s.	n.s.	n.s.
Respirator	Basal	0	0	1 (3.6)	0
(*n*, %)	After 1 week	0	9 (11)	6 (21)	5 (62)
	Δ (%)	0	NQ	500	NQ
	*p* value	n.s.	**<0.0001**	**<0.0001**	**<0.0001**
Intubation	Basal	0	0	0	0
(*n*, %)	After 1 week	0	9 (11)	4 (14)	5 (62)
	Δ (%)	0	NQ	NQ	NQ
	*p* value	n.s.	**<0.0001**	**<0.0001**	**<0.0001**
Platelets	Basal	306 (232; 386)	356 (278; 456)	432 (318; 457)	395 (189; 564)
(10^3^/mmc)	After 1 week	332 (265; 372)	330 (287; 382)	375 (319; 423)	348 (236; 589)
(150–450)	Δ (%)	−3.0 (−10; 7)	−6.0 (−15; 1.0)	−6.0 (−16; 4.5)	−1.5 (−8.0; 4.5)
	*p* value	n.s.	**<0.0001**	**0.005**	n.s.
WBC (N/mmc)	Basal	5910 (4270; 9960)	8550 (5960; 10,950)	8560 (7680; 11,545)	4920 (4282; 6400)
(4500–11,000)	After 1 week	6980 (5325; 9000)	8170 (6230; 9850)	9200 (7425; 12,640)	9555 (6332; 14,380)
	Δ (%)	20 (−11; 40)	−2.4 (−19; 27)	6.1 (−17; 39)	77 (11; 137)
	*p* value	n.s.	n.s.	n.s.	n.s.
Neutrophils (%)	Basal	70 (60; 75)	77 (71; 82)	78 (66; 84)	73 (65; 83)
(40–75)	After 1 week	66 (56; 72)	73 (60; 79)	69 (63; 78)	82 (74; 88)
	Δ (%)	−6.7 (−16; 0.7)	−6.5 (−17; 0.0)	−3.6 (−11; 4.1)	14 (−5.6; 21)
	*p* value	**0.030**	**<0.0001**	n.s.	n.s.
Neutrophils	Basal	4567 (2547; 5873)	6453 (4524; 8334)	6847 (5060; 8837)	3315 (2721; 5585)
(N/mmc)	After 1 week	4578 (3062; 5885)	5407 (4111; 7732)	7018 (4715; 8659)	7560 (3758; 12,830)
(2000–8000)	Δ (%)	2.1 (−22; 38)	−12 (−32; 25)	8.2 (−12; 38)	75 (2.8; 176)
	*p* value	n.s.	n.s.	n.s.	**0.050**
Lymphocytes	Basal	20 (12; 28)	12 (8.0; 18)	11 (8.5; 20)	17 (8.3; 23)
(%)	After 1 week	23 (18; 33)	16 (12; 25)	14 (10; 24)	10 (4.5; 15)
(20–40)	Δ (%)	25 (−9; 64)	33 (−12; 114)	20 (−11; 75)	−33 (−71; 75)
	*p* value	n.s.	**<0.0001**	n.s.	n.s.
Lymphocytes	Basal	1160 (621; 1858)	988 (614; 1500)	991 (634; 1783)	780 (419; 1222)
(N/mmc)	After 1 week	1663 (1195; 2185)	1256 (904; 1819)	1265 (939; 2005)	952 (350; 1340)
(1500–3000)	Δ (%)	31 (3.6; 97)	19 (−16; 106)	27 (−4.9; 116)	−15 (−57; 214)
	*p* value	**0.003**	**<0.0001**	**0.028**	n.s.
NLR (ratio)	Basal	3.5 (2.2; 6.3)	6.3 (3.8; 10)	6.9 (3.5; 9.7)	4.3 (2.8; 12)
	After 1 week	3.0 (1.7; 4.0)	4.3 (2.5; 6.6)	5.2 (2.7; 7.7)	8.8 (5.1; 20)
	Δ (%)	−30 (−49; 9.9)	−32 (−63; 11)	−21 (−50; 15)	84 (−40; 357)
	*p* value	**0.023**	**0.001**	n.s.	n.s.
Ferritin (ng/mL)	Basal	267 (241; 422)	427 (280; 545)	540 (390; 638)	484 (264; 660)
(20–250)	After 1 week	274 (220; 407)	367 (249; 487)	491 (359; 601)	472 (267; 650)
	Δ (%)	−8.0 (−13; −4.5)	−7.0 (−12; −1.2)	−9.0 (−11; −4.7)	−1.0 (−10; 6.2)
	*p* value	**<0.0001**	**<0.0001**	**<0.0001**	n.s.
C-reactive	Basal	19 (12; 46)	45 (25; 68)	63 (36; 78)	50 (42; 64)
protein (mg/L)	After 1 week	5.0 (4.0; 10)	11 (6.0; 18)	19 (9.5; 61)	82 (16; 165)
(0–5)	Δ (%)	−69 (−83; −41)	−75 (−83; −61)	−72 (−78; −17)	72 (−82; 171)
	*p* value	**<0.0001**	**<0.0001**	**0.050**	n.s.

Reference ranges are reported for hematology and clinical chemistry parameters. Paired comparisons between the data at hospital admission (basal) and after 1 week were evaluated by Wilcoxon signed-rank test. Significant values are reported in bold. NLR: neutrophils/lymphocytes ratio; n.s.: not significant; N.Q.: not quantifiable; WBC: white blood cells.

**Table 2 jcm-11-04306-t002:** Comparison of serum ILs (pg/mL) in 144 COVID-19 patients with different severity at hospital admission (basal) and after 1 week. Median (interquartile range). For each parameter, we report the reference range.

		Mild (*n* = 29)	Moderate (*n* = 79)	Severe (*n* = 28)	Died (*n* = 8)
IL-6	Basal	40 (26; 129)	133 (37; 275)	27 (21; 36)	411 (241; 3288)
(0.0–4.5)	After 1 week	30 (23; 66)	59 (24; 170)	33 (27; 90)	1287 (157; 3367)
	Δ (%)	−23 (−67; 5.7)	−29 (−75; 4.0)	3.2 (−37; 157)	−34 (−42; 570)
	*p* value	**0.031**	**0.043**	n.s.	n.s.
IL-10	Basal	7.8 (5.5; 12.2)	9.3 (5.4; 19.4)	2.8 (0.1; 7.8)	9.6 (6.5; 23)
(0.0–3.0)	After 1 week	7.3 (5.4; 8.5)	6.7 (4.6; 9.2)	7.6 (4.5; 24)	12.7 (6.9; 32)
	Δ (%)	−5.5 (−38; 4.4)	−31 (−53; −8.7)	54 (−6.0; 193)	−3.1 (−42; 103)
	*p* value	n.s.	**0.002**	n.s.	n.s.
IL-17A	Basal	2.0 (2.0; 2.3)	3.2 (2.0; 5.3)	3.5 (2.0; 8.5)	2.9 (2.2; 3.4)
(0.0–2.0)	After 1 week	2.0 (2.0; 4.5)	3.2 (2.0; 7.7)	6.0 (2.0; 12)	2.1 (2.0; 25)
	Δ (%)	0.0 (0.0; 31)	0.0 (−27; 34)	68 (−40; 301)	−8 (−39; 731)
	*p* value	n.s.	n.s.	n.s.	n.s.

Paired comparisons between the data at hospital admission (basal) and after 1 week were evaluated by Wilcoxon signed-rank test. Significant values are reported in bold. IL: interleukin; n.s.: not significant.

**Table 3 jcm-11-04306-t003:** Comparison of ILs (pg/mL) and circulating lymphocytes (N/mmc) between severe/dying COVID-19 patients, treated and not treated with steroids before hospitalization. We report the data at hospital admission (basal) and after 1 week. Median (interquartile range). For each parameter, we report the reference range.

		Not Treated (*n* = 11)	Treated (*n* = 24)	*p* Value (a)
IL-6	Basal	42.2 (31.2–403)	24.3 (20.0–34.4)	**0.008**
(0.0–4.5)	After 1 week	106 (32.8–2315)	74.4 (25.5–207)	n.s.
	*p* value (b)	n.s.	n.s.	
IL-10	Basal	9.7 (4.8–57.1)	1.7 (0.1–8.2)	**0.015**
(0.0–3.0)	After 1 week	23.1 (6.5–37.9)	6.9 (6.8–7.0)	n.s.
	*p* value (b)	n.s.	n.s.	
IL-17A	Basal	2.5 (2.0–3.4)	2.8 (2.0–3.1)	n.s.
(0.0–2.0)	After 1 week	3.3 (2.0–12.6)	2.0 (2.0–2.1)	n.s.
	*p* value (b)	n.s.	n.s.	
Total lymphocytes	Basal	785 (388–1241)	991 (639–1794)	n.s.
(1500–3000)	After 1 week	926 (390–1776)	1358 (1063–2005)	n.s.
	*p* value (b)	n.s.	**0.035**	
NLR (%)	Basal	4.0 (2.5–10.5)	6.7 (3.5–10.3)	n.s.
	After 1 week	6.6 (2.6–21.5)	5.6 (3.0–7.7)	n.s.
	*p* value (b)	n.s.	n.s.	
Naïve	Basal	408 (251–900)	614 (346–1353)	n.s.
(126–1121)	After 1 week	666 (182–977)	876 (497–1152)	n.s.
	*p* value (b)	n.s.	n.s.	
Memory	Basal	262 (145–546)	350 (230–693)	n.s.
(319–1184)	After 1 week	268 (98–567)	535 (324–733)	n.s.
	*p* value (b)	n.s.	**0.042**	
T cytotoxic	Basal	235 (49–422)	234 (129–405)	n.s.
(229–1112)	After 1 week	131 (105–267)	328 (239–517)	n.s.
	*p* value (b)	n.s.	**0.009**	
B	Basal	50 (36–165)	125 (68–265)	n.s.
(72–520)	After 1 week	98 (31–401)	154 (98–303)	n.s.
	*p* value (b)	n.s.	**0.008**	
Total	Basal	85 (51–207)	169 (119–358)	n.s.
activated	After 1 week	117 (56–442)	213 (136–349)	n.s.
(86–799)	*p* value (b)	n.s.	**0.005**	
T activated	Basal	15.8 (9.0–27.9)	17.1 (7.9–40.3)	n.s.
(14–411)	After 1 week	16.8 (12.7–23.5)	31.7 (17.3–57.7)	n.s.
	*p* value (b)	n.s.	**0.028**	
TH17	Basal	24.3 (11.8–67.2)	32 (19–68)	n.s.
(3.8–60.0)	After 1 week	18.5 (10.0–42.2)	71 (44–86)	**0.008**
	*p* value (b)	n.s.	**<0.0001**	

Paired comparisons between the data at hospital admission (basal) and after 1 week were evaluated by Wilcoxon signed-rank test (*p*-value (b). Comparisons between treated and non treated patients were evaluated by Mann-Whitney U-test (*p*-value (a)). Significant values are reported in bold. n.s.: not significant.

## Data Availability

The data presented in this study are available on request from the corresponding author.

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
