# Peer review of "Lymphocyte Population Changes at Two Time Points during the Acute Period of COVID-19 Infection"

_jcm, 2022, doi:10.3390/jcm11154306_

Round 1

Reviewer 1 Report

Some comments for authors

-Introduction: Revise your abbreviations (eg, NK cells you should refer to the term (natural killer) at firm time

-In methods:

*Spelling errors are found (eg; nasopharyngeal) kindly revise the whole manuscript.

*You gave (steroids) for 80 patients but you should specify here what type of steroids you used.

*Regarding died patients, you should mention the time period of death and if any patient died before one week of 1st sample collection?

- In Results: P-value in table 5 is not quite clear, revise please

- In Discussion:

*“We previously observed the increase of serum ILs and the reduction of most lymphocyte populations in blood from hospitalized COVID-19 patients, either during the 1st [9] and during the 2nd [13] pandemic wave” You should write pandemic waves in Italy.

*” NK cells, that in turn are inhibited by cytokines [18], resulted also reduced in most patients at..” Grammar revision.

* “These data match with the trend of serum interleukins, indicating a vicious circle in severe COVID-19 patients between cytokine production and TH lymphocyte activation, which slows down disease remission” Here you should cite a related recent reference, I recommend this one: https://doi.org/10.3390/pathogens10050565 )

Author Response

Dear Reviewer 1,

We would like to thank you for the time spent revising our manuscript. Please see below the point-by-point reply to your comments.

-Introduction: Revise your abbreviations (eg, NK cells you should refer to the term (natural killer) at firm time

Response: We have added natural killers in the introduction and referred to the as NK (line 48) in the manuscript

-In methods:

*Spelling errors are found (eg; nasopharyngeal) kindly revise the whole manuscript.

Response: We have corrected the typos in the methods and revised all manuscript

*You gave (steroids) for 80 patients but you should specify here what type of steroids you used.

Response: In the manuscript, we referred to a steroid therapy received before hospital admission. All treated patients received dexamethasone. We have added this information to the manuscript. Please see line 62.

*Regarding died patients, you should mention the time period of death and if any patient died before one week of 1st sample collection?

Response: No patient died before 1-week collection. The average day of death was 11 days. Please see lines 127-129.

- In Results: P-value in table 5 is not quite clear, revise please

Response: As suggested, we have revised the p-value in the new table 3. Specifically, we added a p-value (a) for the comparisons between treated and non treated patients, and a p-value (b) for the comparisons between the data at hospital admission (basal) and after 1 week.

- In Discussion:

*“We previously observed the increase of serum ILs and the reduction of most lymphocyte populations in blood from hospitalized COVID-19 patients, either during the 1st [9] and during the 2nd [13] pandemic wave” You should write pandemic waves in Italy.

Response: We have added the suggested correction

*” NK cells, that in turn are inhibited by cytokines [18], resulted also reduced in most patients at..” Grammar revision.

Response: We have corrected and revised all manuscript

* “These data match with the trend of serum interleukins, indicating a vicious circle in severe COVID-19 patients between cytokine production and TH lymphocyte activation, which slows down disease remission” Here you should cite a related recent reference, I recommend this one: https://doi.org/10.3390/pathogens10050565 )

Response: We have added the suggested reference (see reference 26)

Reviewer 2 Report

In the manuscript by Scalia et al, the authors aim to evaluate changes in serum IL levels and lymphocytes at and a week after hospitalization. Authors report increase in ILs and reduction in lymphocyte population at the onset that tend to normalize after a week of hospitalization; correlating to the severity of disease, these markers remain elevated in severe patients. IL6 has been shown to be a prognosticator for Covid severity according to several studies reported in the literature. The present manuscript fails to stand out to present any new findings than already reported in the literature.

Again with respect to B lymphocyte levels that increased a week after hospitalization in severe diseased patients that died eventually, a mechanistic pathway is unknown.

Authors should also improve the quality of their figures, as most of them are pixelated and very hard to read.

Author Response

Dear Reviewer 2,

We would like to thank you for the time spent revising our manuscript. Please see below the point-by-point reply to your comments.

Comments and Suggestions for Authors

In the manuscript by Scalia et al, the authors aim to evaluate changes in serum IL levels and lymphocytes at and a week after hospitalization. Authors report increase in ILs and reduction in lymphocyte population at the onset that tend to normalize after a week of hospitalization; correlating to the severity of disease, these markers remain elevated in severe patients. IL6 has been shown to be a prognosticator for Covid severity according to several studies reported in the literature. The present manuscript fails to stand out to present any new findings than already reported in the literature.

Again with respect to B lymphocyte levels that increased a week after hospitalization in severe diseased patients that died eventually, a mechanistic pathway is unknown.

Response: Since the beginning of the Sars-CoV-2 pandemic, many efforts have been made to understand the physiological implication of this multiorgan infection and has been shown that phenotypes can also vary in different populations. Therefore, it has been of great importance that the same evidence had been confirmed in different ethnic populations, at different patient’s ages, and at different stages of the disease while adding new findings that could help to better understand this infection. 

Here as the reviewer commented, in our population, we first show that COVID-19 subjects presented with an increase in interleukin levels and a reduction of most lymphocytes’ populations at the time of admission. Although this information has been already shown by our group and others, we think it was extremely important to report in order to characterize our population. Furthermore, before hospitalization, many patients started the steroids treatment at home while others didn't. This has made the assessment of inflammatory markers and lymphocyte counts important to study disease progression, especially in severe patients.

In fact, to the best of our knowledge, we show for the first time that at least in severe patients, treatment with steroids before hospitalization had a significant impact on disease development. As shown in Table 3, at hospital admission, treated patients had significantly lower levels of IL-6 and IL-10 as compared to untreated patients. In addition, after one week of hospitalization, we observed in treated patients a significant increase in total, memory, T cytotoxic, B, total activated, T activated, and TH17 lymphocytes as compared to untreated patients, while no significant differences were observed in untreated patients. Most importantly, we report that all dead patients were the ones that did not receive any steroids before hospitalization.  Furthermore, here we report the novel finding (at least at the time of submission) that in severe and dying patients, a reduced level of B cells at hospitalization increased after one week of hospitalization.

Finally, although a limited period follow-up has been presented here, we believe the evidence presented will strengthen our comprehension of SARS-CoV-2 infection, and at the same time add new findings such as the importance of starting the steroids treatment at home, especially in the weakest subjects that could progress to a severe condition.   

Authors should also improve the quality of their figures, as most of them are pixelated and very hard to read.

Response: We have improved the quality of the figures

Reviewer 3 Report

Scalia G et al. describe the evolution (baseline versus 1 week) of blood immunological markers (cellular: extended lymphocyte phenotyping and soluble factors: cytokines) during symptomatic forms of COVID-19. The inclusion of the 144 patients was carried out during the 1st and 2nd wave of the epidemic in Italy. The authors found a decrease in circulating pro-inflammatory cytokines and an increase in the level of circulating lymphocytes between the baseline and the 1st week. Their previous work had already focused on similar observations, the originality of this work is to compare the markers between the baseline and the 1st week.

However, this work deserves major revisions before publication.

1) These results are not original. Lymphopenia has already been well described and linked to prognosis in severe forms of COVID-19 (reference 9 for example). This work does not provide any major information.

2) The population is poorly described: what are mild, moderate and severe forms of COVID-19? Wouldn't it be more interesting to classify patients according to the WHO respiratory scale? The lack of precision on this classification makes it impossible to interpret the results of this study.

Other important concerns are:

3) The bibliography needs to be improved

The most notable references in the literature are not cited.

For example, regarding references on lymphocyte markers:

Bergamaschi L et al. Immunity. 2021 Jun 8;54(6):1257-1275.e8

Kreutmair S, et al. Immunity. 2021 Jul 13;54(7):1578-1593.e5

Melms JC et al. Nature. 2021 Jul;595(7865):114-119

Another example for NK cell references

Witkowski M et al. Nature. 2021 Dec;600(7888):295-301

Kramer B et al. Immunity. 2021 Nov 9;54(11):2650-2669.e14

Some references are not suitable. For example, reference 5 does not justify what is in the text.

4) The study recently carried out by the same team (reference 13) included 274 patients from the 2nd wave and 35 patients from the 1st wave. Why were not all patients included in this study? Is this missing data? Is it really the same cohort? A figure with a flow-chart seems essential to know how many patients have been excluded and why.

5) Methods: results are given in the method, in particular the number of patients included. This part must appear in results. It is absolutely necessary to specify the definition of mild, moderate and severe COVID. Gating strategies on flow cytometry should be specified here. The figure is not required or appear in supplementary material.

6) Results (more figures would be needed to present biological results, the tables are difficult to read):

a. Table 1: a simplification is necessary, with more relevant clinical data: WHO scale +++. What does "Respirator" mean? The NLR is not a percentage but a ratio.

b. The absence of statistical difference between baseline and week 1 in patients who died is sometimes surprising: NLR, CRP and WBC? Are the authors sure of their calculations?

c. Authors should specify numbers and p values ​​in the text.

d. Table 5 is certainly the most interesting result and to highlight. On the other hand, why are there only 35 patients when the initial patients either received or did not receive corticosteroids? In the text Line 85-86, it is specified: “Among the 144 patients, 80 (55%) were treated with steroids before hospitalization, while 64 (45%) were not treated”.

It should also be specified which corticosteroid they received and for how long.

e. Why only 8 patients in Figure 3?

7) The discussion and the conclusion largely depend on the patients being compared, which the authors must specify.

Author Response

Dear Reviewer 3,

We would like to thank you for the time spent revising our manuscript. Please see below the point-by-point reply to your comments.

Comments and Suggestions for Authors

Scalia G et al. describe the evolution (baseline versus 1 week) of blood immunological markers (cellular: extended lymphocyte phenotyping and soluble factors: cytokines) during symptomatic forms of COVID-19. The inclusion of the 144 patients was carried out during the 1st and 2nd wave of the epidemic in Italy. The authors found a decrease in circulating pro-inflammatory cytokines and an increase in the level of circulating lymphocytes between the baseline and the 1st week. Their previous work had already focused on similar observations, the originality of this work is to compare the markers between the baseline and the 1st week.

However, this work deserves major revisions before publication.

1) These results are not original. Lymphopenia has already been well described and linked to prognosis in severe forms of COVID-19 (reference 9 for example). This work does not provide any major information.

Response: As the reviewer is pointing out, in our manuscript we first show that COVID-19 subjects had an increase in cytokines levels and lymphopenia at the time of hospital admission. Although we are aware that this information has been already shown by our group and others, we think it was extremely important to report in order to characterize our population. In fact, during this pandemic, therapies have been continuously modified and, in this work, we had many patients that started the steroids treatment before hospitalization while others didn't. Therefore, to study if the house treatment could have had an impact on disease progression, we thought that the evaluation of inflammatory markers and lymphocyte counts would have been necessary.  

In fact, here we show that in severe patients steroids treatment before hospitalization positively impacts disease progression. In Table 3 it could be seen that already at admission time treated patients had significantly lower levels of IL-6 and IL-10 as compared to untreated patients. Furthermore, after one week of hospitalization, treated patients had a significant increase in total, memory, T cytotoxic, B, total activated, T activated, and TH17 lymphocytes as compared to untreated patients, while no significant differences were observed in untreated patients. Moreover, and most importantly, we report that all dead patients were the ones that did not receive any steroids before hospitalization.  Finally, here we report the novel finding (at least at the time of submission) that in severe and dying patients, a reduced level of B cells at hospitalization increased after one week of hospitalization.

2) The population is poorly described: what are mild, moderate and severe forms of COVID-19? Wouldn't it be more interesting to classify patients according to the WHO respiratory scale? The lack of precision on this classification makes it impossible to interpret the results of this study.

Response: Our classification was already based on WHO guidelines. As suggested by the reviewer we have described in material and methods the WHO classification that has been used to stratify patients in mild, moderate and severe.

Other important concerns are:

3) The bibliography needs to be improved. The most notable references in the literature are not cited. For example, regarding references on lymphocyte markers:

Bergamaschi L et al. Immunity. 2021 Jun 8;54(6):1257-1275.e8

Kreutmair S, et al. Immunity. 2021 Jul 13;54(7):1578-1593.e5

Melms JC et al. Nature. 2021 Jul;595(7865):114-119

Another example for NK cell references

Witkowski M et al. Nature. 2021 Dec;600(7888):295-301

Kramer B et al. Immunity. 2021 Nov 9;54(11):2650-2669.e14

Some references are not suitable. For example, reference 5 does not justify what is in the text.

Response: As suggested by the reviewer we have added the reference.

4) The study recently carried out by the same team (reference 13) included 274 patients from the 2nd wave and 35 patients from the 1st wave. Why were not all patients included in this study? Is this missing data? Is it really the same cohort? A figure with a flow-chart seems essential to know how many patients have been excluded and why.

Response: We have included in this study 32 patients from the 1st wave and 112 patients from the 2nd wave, for which we had a blood sample after 1 week hospitalization. We added a flow-chart of patients’ selection in Supplementary Material as Figure S1.

5) Methods: results are given in the method, in particular the number of patients included. This part must appear in results. It is absolutely necessary to specify the definition of mild, moderate and severe COVID. Gating strategies on flow cytometry should be specified here. The figure is not required or appear in supplementary material.

Response: As suggested, we have moved the patient description in the result section. Furthermore, in material and methods we have specified the definition of mild, moderate and severe according to WHO guidelines. Finally, we have added in the methods section the gating strategy and moved in supplementary material as Figure S1.

6) Results (more figures would be needed to present biological results, the tables are difficult to read):

Response: We added Figure 2 and Figure 3 that report the numbers of lymphocyte subpopulations of old Table 3 and Table 4. These two tables have been moved in Supplementary material.

  1. Table 1: a simplification is necessary, with more relevant clinical data: WHO scale +++. What does "Respirator" mean? The NLR is not a percentage but a ratio.

Response: As suggested by the reviewer we have simplified table 1 by leaving only the more relevant clinical data. Furthermore, as already mentioned earlier, we have added in the material and methods the WHO classification, but left in the text only the mild, moderate and severe classification because we think it is easier for the reader to understand. Respirator is referred to the use of an oxygen mask and we have specified it in the Table 1 legend. Finally, we have corrected with RATIO the NLR value in table 1.

  1. The absence of statistical difference between baseline and week 1 in patients who died is sometimes surprising: NLR, CRP and WBC? Are the authors sure of their calculations?

Response: We understand that it may seems “logic” that in the patient who died there should be an increase in NLR ratio, CRP and WBC. As can be seen in table 1 we do see an increase in these parameters, but it was not significative because of the high variability also due to the small number of patients in this group (8 patients). We have analyzed these data multiple times and we are sure of the statistical analysis.  

  1. Authors should specify numbers and p values ​​in the text.

Response: As suggested we added the numbers and p-values in the text.

  1. Table 5 is certainly the most interesting result and to highlight. On the other hand, why are there only 35 patients when the initial patients either received or did not receive corticosteroids? In the text Line 85-86, it is specified: “Among the 144 patients, 80 (55%) were treated with steroids before hospitalization, while 64 (45%) were not treated”.

Response: In the new table 3 we only report the data for severe and died group (35 patients). We had to exclude one patient because we did not have information about the treatment before hospitalization.

It should also be specified which corticosteroid they received and for how long.

Response: All patients receive dexamethasone before hospital admission for 2-3 days (lines 130-131).

 Why only 8 patients in Figure 3?

Response: In the new Figure 4 we analyzed only the group of patients who died (n = 8).

7) The discussion and the conclusion largely depend on the patients being compared, which the authors must specify.

Response: As suggested we have specified that our discussion and conclusion depend on patients being compared.

Round 2

Reviewer 2 Report

The revisions are satisfactory. The manuscript however needs thorough proof reading for grammatical and typographical errors.

Reviewer 3 Report

I thank the authors for their work and changes.

Treatment by dexamethasone should appear in Table 1.